# Evaluation of the reduction in perceived and performance fatigability by functional compression tights during squat exercises via electromyography and electroencephalography analysis

Takuma Isshiki◎, Shinnosuke Tsukada◎, Minami Akao, Yuna Ishikura, Hayato Shigetoh◎*, Takayuki Kodama◎, Junya Miyazaki

Department of Physical Therapy, Faculty of Health Science, Kyoto Tachibana University, Kyoto, Japan

◎ These authors contributed equally to this work.
* shigeto@tachibana-u.ac.jp

## Abstract

### Objectives

Functional compression tights are widely used to support muscle activity, enhance blood flow and reduce fatigue, which comprises performance (motor or cognitive) and perceived fatigability. Although previous studies have reported their effects on motor performance fatigability, little is known about their effects on cognitive performance fatigability or brain activity. This study aimed to evaluate quantitatively and comprehensively the effects of functional compression tights on perceived fatigability, muscle activity, and electroencephalographic (EEG) responses.

### Methods

Twenty healthy young adults performed squat tasks under two conditions (with and without functional compression tights) using a crossover design. Muscle activity was measured using surface electromyography (EMG) of the five thigh muscles to calculate the root mean square (RMS) and mean frequency (MF). EEG activity was recorded in the Fp1, Fp2, and Fz regions, and power change rates in the alpha, beta, and theta bands were calculated. Perceived fatigability was assessed using a numerical rating scale. Wilcoxon signed-rank tests were used for between-condition comparisons, and Spearman's rank correlation coefficients were calculated to examine the comprehensive relationships among perceived fatigability, muscle activity, and EEG activity.

### Results

Wearing tights significantly reduced perceived fatigability ($p=0.003$) and RMS of the vastus medialis obliquus (VMO) ($p=0.041$). Although not statistically significant

**Data availability statement:** All relevant data are within the manuscript and its Supporting Information files.

**Funding:** This research was supported by a collaborative research grant from Wacoal Corp. and by a Grant-in-Aid for Scientific Research (KAKENHI) from the Japan Society for the Promotion of Science (JSPS) [Grant Number 25K141790].

**Competing interests:** The authors have declared that no competing interests exist.

($p = 0.054$), the moderate effect size suggests a stabilizing effect of functional compression tights on frontal EEG activity. Under the condition with functional compression tights, the RMS of the RF positively correlated with perceived fatigability ($\rho = 0.53$), while MF showed a negative correlation ($\rho = -0.70$). Positive correlations were also observed between perceived fatigability and the alpha/theta powers at Fp2 and alpha power at Fz.

## Discussion

Wearing functional compression tights may alleviate motor and cognitive performance fatigability by stabilizing muscle and frontal brain activity. These findings support its utility in managing motor and cognitive fatigability during physical activity.

---

### Introduction

Functional compression tights are designed to support muscles and reduce fatigue during exercise through mechanical compression. Wearable taping tights mimic athletic taping to suppress muscle oscillation and enhance joint stability, while standard compression tights apply uniform pressure to promote blood circulation and alleviate fatigue [1–3]. However, their effects may vary depending on individual characteristics, exercise type, and garment application. Some studies have reported inconsistent benefits or even adverse effects such as restricted movement or altered muscle activation [4–6]. Additionally, most research focuses on short-term effects, and the long-term efficacy and safety of these garments remain unclear [7,8].

Fatigue is currently conceptualized as a psychophysiological condition that arises during or following motor or cognitive tasks and is defined by two interrelated components: performance fatigability and perceived fatigability [9,10]. Performance fatigability refers to a measurable decline in task performance, which can be further categorized into motor performance fatigability—such as a reduction in muscular force output—and cognitive performance fatigability—such as decreased accuracy, reaction speed, or attentional stability in cognitive tasks. In contrast, perceived fatigability reflects the subjective experience of fatigue, including sensations of effort, discomfort, and reduced motivation. These two components are interdependent but influenced by distinct physiological and psychological mechanisms, including neuromuscular function, sensory and affective feedback, cognitive control, and overall homeostatic state.

Motor performance fatigability can be objectively quantified using surface electromyography (EMG), which captures myoelectric changes in both time and frequency domains. In the time domain, root mean square (RMS) reflects the amplitude of muscle activation, typically increasing in early motor performance fatigability due to greater motor unit recruitment, then decreasing under severe motor performance fatigability. In the frequency domain, mean frequency (MF) generally decreases with progressive motor performance fatigability as muscle fiber conduction velocity slows. RMS and MF are therefore considered complementary indicators of motor

performance fatigability [11]. Previous studies suggest that functional compression tights can reduce RMS and attenuate the decline in MF, indicating lower muscle activation and mitigated motor performance fatigability progression [5,12]. These findings highlight the importance of assessing muscle motor performance fatigability using both amplitude- and frequency-based EMG measures to capture the multifaceted nature of motor performance fatigability.

Central nervous system factors are also closely associated with fatigue, with central neural activity influencing peripheral muscle responses [11]. In motor performance fatigability, the depletion of muscular capacity affects brainwave activity. As cognitive performance fatigability progresses, reductions and irregularities in alpha wave activity have been reported in the frontal regions (Fp1, Fp2, Fz) during and after exercise [13]. According to a previous study [14], the left frontal area (Fp1) is associated with positive emotional states, whereas the right frontal area (Fp2) is linked to negative emotional processes. Additionally, beta activity in the frontal cortex, especially in the Fz region, increases in response to muscular tension and psychological stress, indicating heightened cognitive load and cognitive performance fatigability progression [15,16]. In cognitive performance fatigability, increased cognitive demands and sustained attention have been shown to influence electroencephalography (EEG) activity. Specifically, elevated alpha activity in the frontal regions (Fp1, Fp2, Fz) has been documented as a compensatory response to attentional decline [17]. Moreover, theta activity in the frontal midline, particularly at Fz, rises as attention and decision-making deteriorate, reflecting greater cognitive workload and cognitive performance fatigability [16,18]. Despite these findings, the effects of functional compression tights on brainwave activity remain poorly understood. Further studies are needed to clarify whether these garments modulate central neural responses to both motor and cognitive performance fatigability.

Although previous studies have reported that wearing functional compression tights can reduce motor performance fatigability—as assessed by muscle activity indices—evidence regarding their influence on brainwave activity remains limited and inconclusive. We hypothesized that the fatigue-attenuating effect of functional compression tights may stem from their combined influence on motor and cognitive performance fatigability. Prior studies have shown that as motor performance fatigability progresses, RMS tend to increase, while MF decreases in EMG signals [11]. Concurrently, reductions in frontal alpha wave activity have been associated with declines in both motor and cognitive performance [13], whereas beta wave activity tends to increase under fatigued conditions [1 5]. Based on these findings, we anticipated that using functional compression tights would mitigate motor performance fatigability, reflected by reduced RMS and preserved or elevated MF. In terms of EEG activity, we further hypothesized that functional compression tights would stabilize brainwave fluctuations—enhancing alpha activity and suppressing beta activity. This study investigated whether changes in muscle activity (i.e., reduced RMS and preserved MF) induced by wearing functional compression tights were associated with variations in EEG-based fatigue indicators cognitive performance fatigability (i.e., alpha, beta, and theta power). By comparing pre- and post-exercise muscle and brain activity, we aimed to quantitatively assess the neuromuscular and neurophysiological effects of functional compression tights. This study sought to provide an integrative evaluation of attenuation of motor and cognitive performance fatigability using both muscular and cortical electrophysiological indicators.

## Participants and methods

### Participants

Twenty healthy young adults (mean age: $20.6 \pm 0.5$ years; 14 males and six females) participated in the study. Individuals with current lower-limb pain or a history of orthopedic disorders affecting the lower extremities were excluded from the study. This study was conducted as part of undergraduate thesis research and was approved by the Research Ethics Committee of Kyoto Tachibana University (Approval No.:24−06). All participants, including minors, provided written informed consent prior to participation in accordance with the principles of the Declaration of Helsinki. Although the study included minors, the ethics committee did not require parental or guardian consent due to the academic nature of the research and the maturity of the participants as university students. The study employed a cross-sectional design. The recruitment and data collection for this study were conducted between 1st June 2024 and 31st August 2024.

## Research protocol

This study employed a crossover design. The participants completed a fatiguing task and associated measurements under two conditions: wearing functional compression tights (CW-X half-type, Wacoal Corp, Kyoto, Japan) and without functional compression tights. A one-week washout period was set between the two experimental sessions. The order of the two conditions was randomly assigned to each participant (Fig 1).

Under each condition, the participants followed a standardized protocol, and muscle activity and EEG signals were recorded. First, the participants maintained a squat posture for 40 seconds to establish pre-fatigue baseline measures. Subsequently, a 2-minute rest period was provided. Subsequently, participants performed repeated squat exercises for the fatiguing task. Immediately after completing the fatiguing task, the participants maintained a squatting posture for 40 seconds to obtain post-fatigue measurements (Fig 2).

## Fatiguing task (squat exercise)

The motor performance fatigability test consisted of repeated squatting. A single squat was defined as a movement sequence starting from a standing position with feet shoulder-width apart and arms crossed in front of the body, descending until the knee joints reached approximately 90 degrees of flexion, and then returning to the upright position. A visual marker was placed at eye level to ensure consistency in squat depth before and after the fatiguing task to serve as a reference point for achieving the target posture. The participants were instructed to perform up to 200 repetitions of the squat movement or to continue until a combination of perceived and motor performance fatigability made it difficult to maintain the proper form. The movement pace was regulated using auditory cues set at 40 beats per minute via a metronome-like beep. The participants were asked to maintain a squat posture at 90 degrees of knee flexion for 40 seconds to evaluate muscle activity under isometric contraction before and after the fatiguing task.

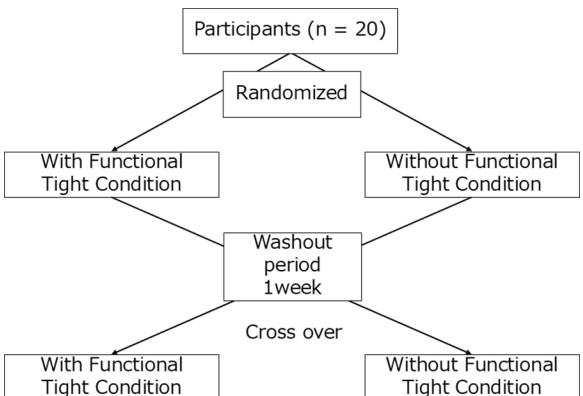

**Fig 1. Research protocol.**

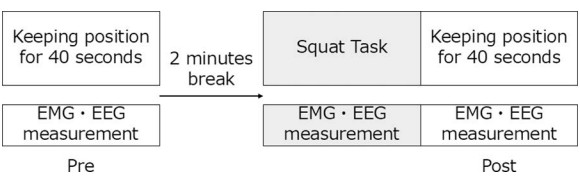

**Fig 2. Assessment protocol.**

## Recording and analysis of muscle activities

Surface EMG data were recorded using a wireless EMG sensor system (BioSignalPlux, Plux, Inc., Lisbon, Portugal). Based on previous research [19], five dominant leg muscles were selected for analysis: the rectus femoris (RF), vastus lateralis (VL), vastus medialis obliquus (VMO), biceps femoris (BF), and semitendinosus (ST). A reference electrode was placed during the radial styloid process. Before electrode placement, the skin at each site was cleaned with alcohol to reduce impedance. Electrode placement followed the recommendations of SENIAM (Surface Electromyography for the Non-Invasive Assessment of Muscles) [20] as follows: RF, midpoint between the anterior superior iliac spine (ASIS), and superior border of the patella. VL: distal one-third of the line between the ASIS and the lateral border of the patella; VMO: distal one-fifth of the line between the ASIS and the anterior margin of the medial collateral ligament at the joint space. BF: midpoint between the ischial tuberosity and the lateral epicondyle of the tibia. ST: Midpoint between the ischial tuberosity and medial epicondyle of the tibia.

EMG signals were sampled at 1000 Hz and preprocessed using a band-pass filter (10–400 Hz). The signals were then full-wave-rectified. Two indicators were calculated to evaluate motor performance fatigability: time (RMS) and frequency domain (MF). For each muscle, a central 10-second segment of isometric contraction data (during squat posture maintenance) was extracted both before (pre) and after (post) the fatiguing task. RMS and MF values were computed for each segment. To account for individual variability in baseline muscle activation, post-task values were normalized relative to pre-task values (Pre = 100%), and the relative change was used for further analysis. This normalization enabled precise comparisons between conditions by eliminating inter-individual differences in the absolute EMG amplitude.

## Recording and analysis of brain wave activities

EEG signals were recorded using an 8-channel wearable EEG device (Altaire; Creact Inc., Japan) at a sampling rate of 1000 Hz. Based on the international 10–20 system, three regions of interest (Fp1, Fp2, and Fz) were selected for analysis due to their established associations with cognitive and motor performance fatigability. The Fp1 and Fp2 regions are involved in cognitive and emotional processing and are known to reflect changes in cognitive performance fatigability and emotional arousal [21]. The Fz region is associated with motor control and motor performance fatigability, making it suitable for evaluating neurophysiological changes following physical exertion [1 7]. This study analyzed alpha waves at Fp1, Fp2, and Fz as indicators of arousal and dearousal states. Alpha wave blocking, or alpha-blocking, is a well-established neurophysiological indicator of cognitive performance fatigability and arousal state. It refers to the suppression of alpha wave activity that typically occurs upon task initiation or exposure to external stimuli, reflecting increased wakefulness and attentional engagement [22]. As cognitive performance fatigability accumulates, this blocking response tends to weaken, leading to a rebound or increase in alpha power [18]. Additionally, beta waves were assessed as markers of attentional arousal and emotional tension [23], while theta waves were evaluated as indicators of cognitive performance fatigability and attentional fluctuation [1 7]. EEG signals were referenced to both earlobes. Dry electrodes were applied to the cleaned forehead area, and the electrode contact was carefully adjusted for optimal signal acquisition. To minimize external influences on EEG activity, the participants were instructed to sleep adequately the night before measurement and refrain from consuming alcohol or caffeine. All recordings were conducted in a quiet controlled environment.

EEG data were preprocessed by applying a bandpass filter (1–30 Hz) to remove the frequency components outside the range of interest. Independent component analysis was used to remove artifacts such as eye blinks and muscle noise, and to isolate neural EEG components. A stable 10-second segment was extracted from the middle of the 40-second isometric squat-holding task performed before and after the fatigue protocol to ensure consistency within the EMG analysis. Time-frequency analysis was conducted using wavelet transformation to calculate the power values in the theta (4–7 Hz), alpha (8–12 Hz), and beta (13–30 Hz) bands. Power values were log-transformed using the natural logarithm. Post-task power values were normalized relative to pre-task values (Pre = 100%) and expressed as a percentage change to account for individual differences.

### Perceived fatigability

To evaluate the perceived fatigability in the lower limbs after the squat task, an 11-point Numerical Rating Scale (NRS) was used. Participants rated their perceived fatigue on a scale ranging from 0 ("no fatigue at all") to 10 ("extremely severe fatigue").

### Statistical analyses

Wilcoxon signed-rank tests were conducted to compare the outcomes between the two conditions: with and without functional compression tights. The variables analyzed included perceived fatigability of the lower limb (NRS), muscle activity indices (RMS and MF), and EEG power changes (alpha, beta, and theta bands) at the Fp1, Fp2, and Fz electrode sites. Spearman's rank correlation coefficients were calculated to examine the relationships between perceived fatigability and physiological indicators (muscle activity and EEG power) and between muscle activity and EEG indices separately for each condition. The significance level was set at $p < 0.05$. All analyses related to the muscle and EEG signals were performed using MATLAB (R2024b, MathWorks, Natick, MA, USA), and statistical analyses were conducted using R software (version 4.2.3).

## Results

### Differences in perceived fatigability between conditions

Perceived fatigability was significantly lower in the functional compression tights condition than that in the without functional compression tights condition ($p = 0.003$, $r = -0.46$) (Fig 3).

### Muscle activity indicators by condition

Regarding the RMS indicator after the fatiguing task, only the VMO showed a significantly lower value with functional compression tights than without functional compression tights conditions ($p = 0.041$, $r = 0.46$). No significant differences were observed between the conditions for the other muscles. In addition, no significant differences were found between conditions for any of the muscles for the MF indicator (Table 1 and Fig 4).

### Differences in EEG activity between conditions

Following the fatiguing task, no significant differences were observed in EEG activity indicators at the Fp1 and Fp2 sites across all frequency bands exhibiting a trend toward smaller effect sizes (Table 2 and Fig 5). At the Fz site, the change in alpha power in the functional compression tights condition was lower than that in the without functional compression tights condition; however, this difference did not reach statistical significance ($p = 0.054$). The functional compression tights condition showed moderate effect sizes for reduced power changes in the alpha ($r = -0.43$), beta ($r = -0.32$), and theta bands ($-0.30$), suggesting a potential stabilizing effect on EEG activity in the frontal midline region.

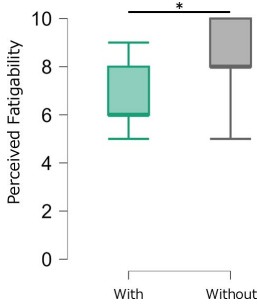

**Fig 3. Comparison of perceived fatigability between conditions.**

**Table 1. Muscle activity indicators by condition.**

| Muscle variable | With Functional Compression Tight | Without Functional Compression Tight | P value | Effect size (r) |
|---|---|---|---|---|
| RMS (%) | | | | |
| RF | 100.4 (100.4) | 122.9 (79.8) | 0.211 | −0.28 |
| VL | 96.4 (83.6) | 84.3 (109.7) | 0.113 | 0.36 |
| VM | 84.7 (78.3) | 148.0 (127.2) | 0.041 | −0.46 |
| BF | 104.8 (185.3) | 120.3 (132.3) | 0.054 | −0.43 |
| ST | 130.3 (53.1) | 130.9 (82.5) | 0.926 | −0.03 |
| MF (%) | | | | |
| RF | 96.4 (7.8) | 95.3 (7.0) | 0.467 | 0.17 |
| VL | 104.8 (12.5) | 109.5 (29.3) | 0.360 | −0.21 |
| VM | 100.6 (13.8) | 96.8 (15.3) | 0.225 | 0.28 |
| BF | 98.2 (11.1) | 98.8 (15.7) | 0.985 | −0.01 |
| ST | 97.1 (7.2) | 99.8 (11.1) | 0.360 | −0.21 |

Data are presented as the median (interquartile range). rectus femoris (RF), vastus lateralis (VL), vastus medialis obliquus (VMO), biceps femoris (BF), and semitendinosus (ST).

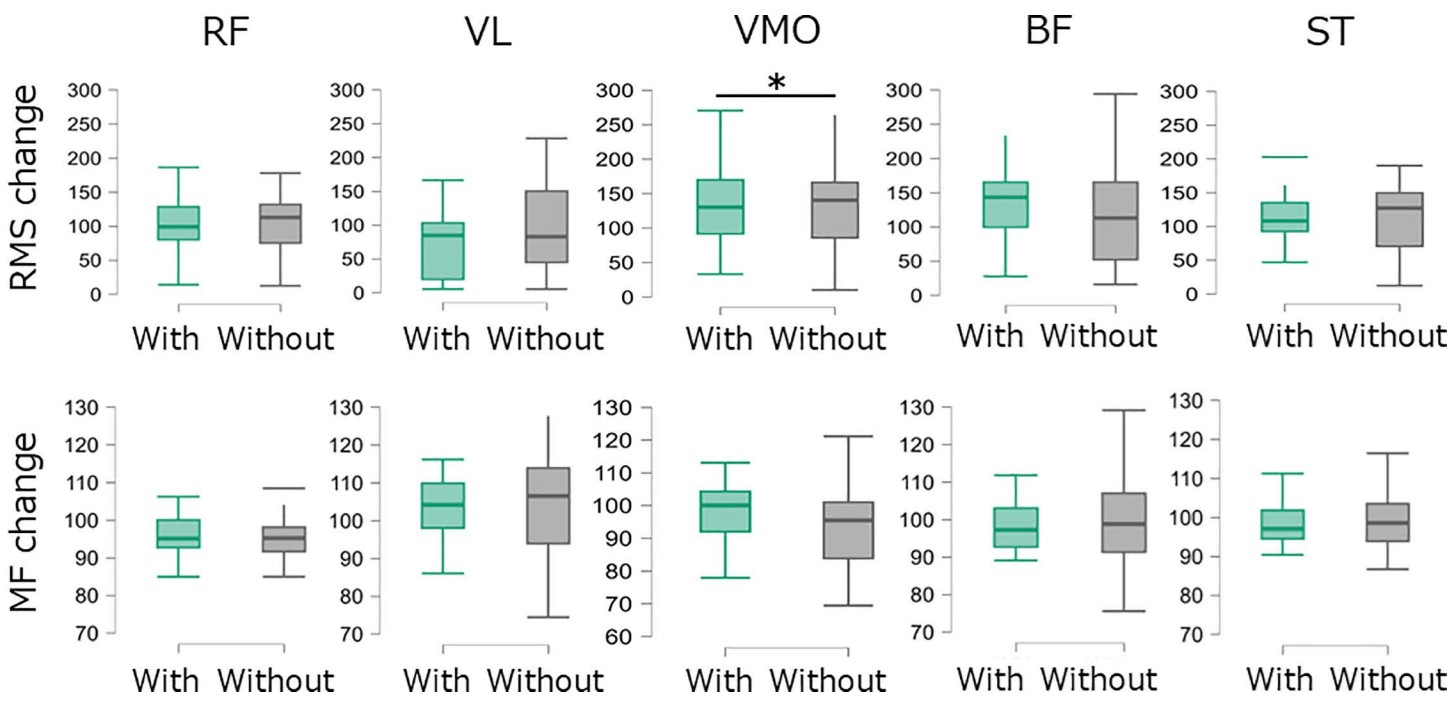

**Fig 4. Box plots of muscle activity indicators by condition.**

## Relationships between perceived fatigability and EMG/EEG indicators

As shown in Fig 6, correlation analyses revealed significant associations between perceived fatigability and EMG indicators. In the condition with functional compression tights, perceived fatigability was positively correlated with the RMS of the RF ($\rho = 0.53$, $p = 0.017$) and negatively correlated with its MF ($\rho = -0.70$, $p = 0.001$). On the other hands, in the condition without functional compression tights, a significant negative correlation was observed between perceived fatigability and the RMS of the VMO ($\rho = -0.50$, $p = 0.024$).

**Table 2. Brain wave activity indicators by condition.**

| Muscle variable | With Functional Compression Tight | Without Functional Compression Tight | P value | Effect size (r) |
|---|---|---|---|---|
| Fp1 | | | | |
| α change (%) | 125.3 (69.2) | 146.1 (47.8) | 0.360 | −0.21 |
| β change (%) | 140.9 (72.9) | 178.3 (107.7) | 0.341 | −0.21 |
| θ change (%) | 117.0 (38.2) | 119.8 (30.4) | 0.563 | −0.13 |
| Fp2 | | | | |
| α change (%) | 144.3 (53.1) | 149.0 (47.6) | 0.514 | −0.15 |
| β change (%) | 154.7 (59.2) | 145.2 (104.4) | 0.926 | 0.02 |
| θ change (%) | 122.2 (19.2) | 120.8 (32.4) | 0.401 | −0.19 |
| Fz | | | | |
| α change (%) | 149.3 (111.0) | 207.4 (82.2) | 0.054 | −0.43 |
| β change (%) | 166.8 (124.0) | 218.8 (159.0) | 0.151 | −0.32 |
| θ change (%) | 146.1 (66.9) | 188.2 (99.2) | 0.185 | −0.30 |

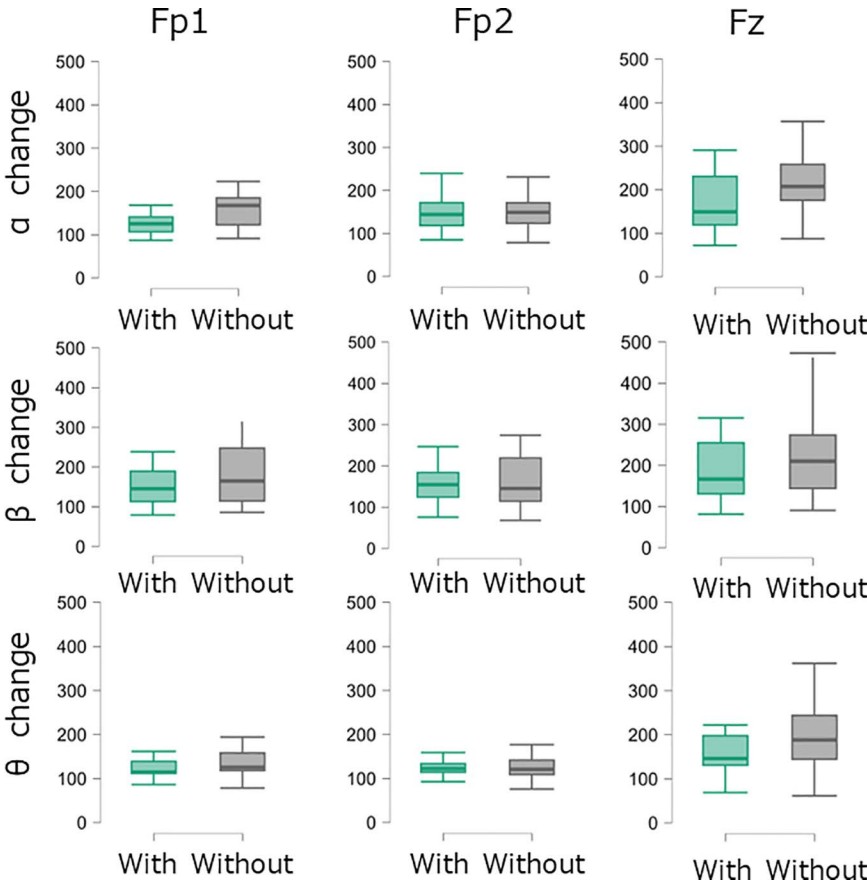

**Fig 5. Box plots of brain wave activity indicators by condition.**

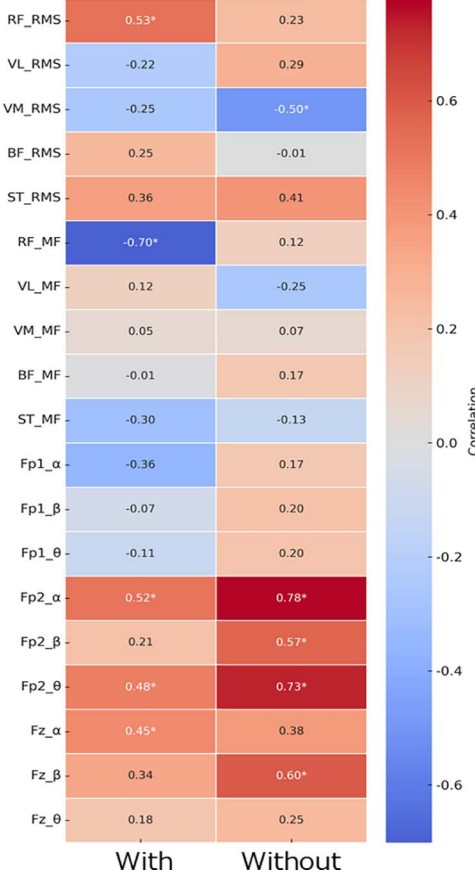

**Fig 6. Correlation heatmaps: Perceived fatigability, EMG, and EEG indicators by condition.**

Regarding correlations with EEG indicators (Fig 6), under the condition with functional compression tights, perceived fatigability showed significant positive correlations with the change in alpha power at Fp2 ($\rho = 0.52$, $p = 0.020$), theta power at Fp2 ($\rho = 0.48$, $p = 0.032$), and alpha power at Fz ($\rho = 0.45$, $p = 0.047$). In the condition without functional compression tights, perceived fatigability was significantly correlated with changes in alpha power ($\rho = 0.78$, $p < 0.001$), beta power ($\rho = 0.57$, $p = 0.009$), and theta power ($\rho = 0.73$, $p < 0.001$) at Fp2, as well as beta power at Fz ($\rho = 0.60$, $p = 0.005$).

### Relationships between muscle activity and EEG indicators

As shown in Fig 7, the correlation analyses in the functional compression tights condition revealed significant associations between muscle activity and EEG power changes. Specifically, the RMS of the RF was positively correlated with beta ($\rho = 0.50$, $p = 0.024$) and theta power ($\rho = 0.51$, $p = 0.020$) at Fp2, as well as beta ($\rho = 0.45$, $p = 0.045$) and theta power ($\rho = 0.54$, $p = 0.013$) at Fz. The MF of the RF showed a significant positive correlation with alpha power at Fp1 ($\rho = 0.59$, $p = 0.006$). Additionally, the RMS of the VL was positively correlated with alpha power at Fp2 ($\rho = 0.51$, $p = 0.021$), and the RMS of the ST was positively correlated with beta power at Fz ($\rho = 0.47$, $p = 0.037$).

On the other hand, in the condition without functional compression tights (Fig 8), significant negative correlations were observed between the RMS of the VMO and alpha ($\rho = -0.63$, $p = 0.003$), beta ($\rho = -0.45$, $p = 0.044$), and theta power ($\rho = -0.58$, $p = 0.007$) at Fp2, as well as beta power at Fz ($\rho = -0.54$, $p = 0.014$).

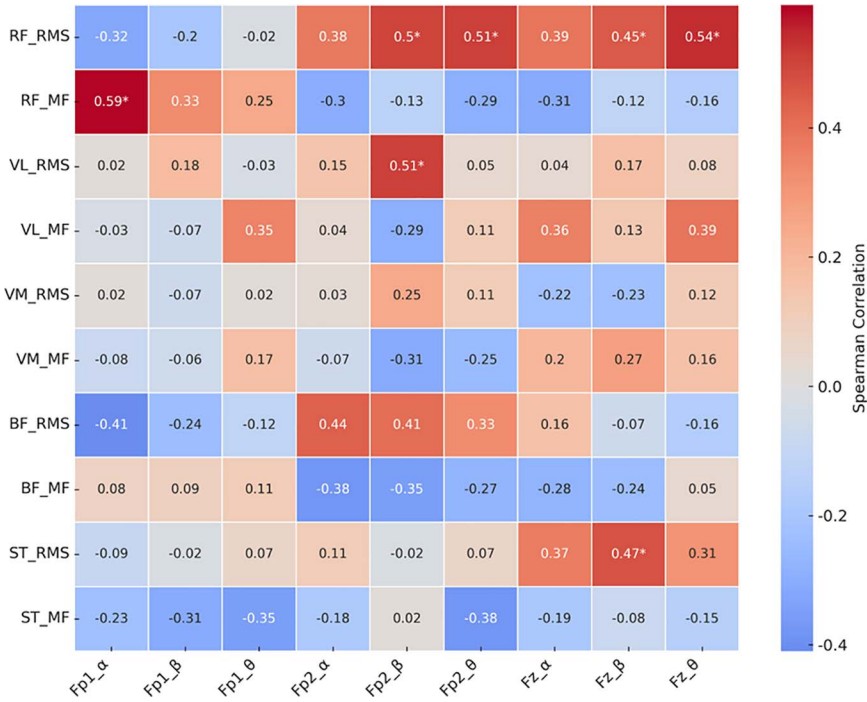

**Fig 7. Heatmap of correlation coefficients between muscle activity and EEG indicators with functional compression tights condition.**

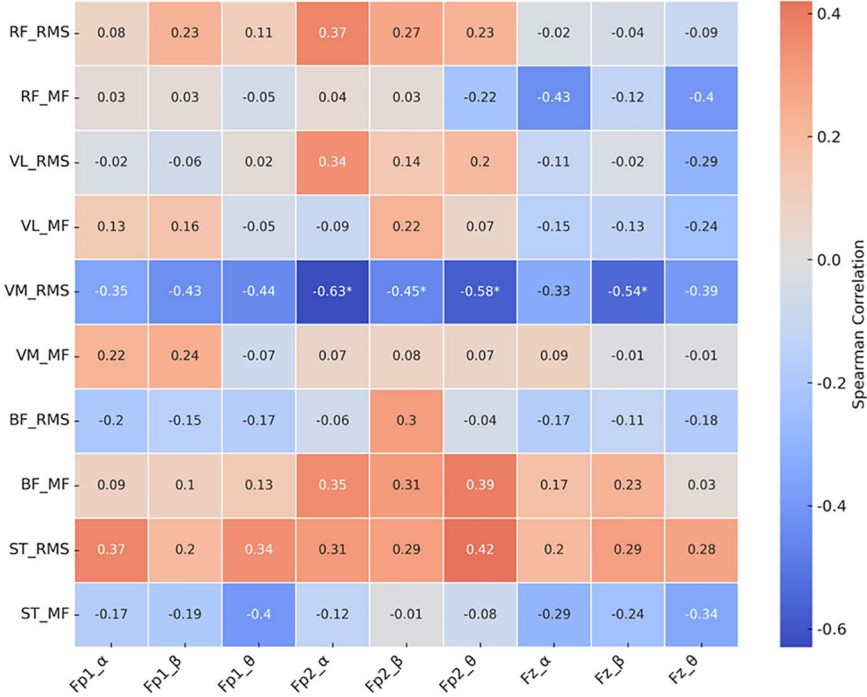

**Fig 8. Heatmap of correlation coefficients between muscle activity and EEG Indicators without functional compression tights condition.**

## Discussion

### Overview of results

This study investigated the effects of functional compression tights on perceived fatigability, muscle activity, and EEG responses. The tights significantly reduced perceived fatigability and selectively suppressed RMS activity in the VMO, suggesting reduced post-task neuromuscular demand. Although no significant changes were observed in EEG across most regions, a relatively stable alpha power in the Fz region may indicate potential modulation of frontal brain activity. These results suggest that compression garments may influence both motor and cognitive performance fatigability through muscle-specific and neurophysiological pathways.

### Effects of compression tights on motor performance fatigability

Wearing functional compression tights reduced perceived fatigability and selectively suppressed EMG activity in the VMO after the fatiguing task. Since RMS typically increases with performance fatigability [11], the lower RMS observed in the VMO suggests reduced neuromuscular demand. This may be due to enhanced joint stability, as the VMO plays a key role in patellar tracking [24]. Prior studies have shown that compression garments can modulate muscle activity based on regional pressure and the alignment of compression bands with muscle fiber direction [25–27]. These factors may explain why no significant effects were found in other muscles like the RF or ST. Regional variation in compression strength and orientation may have limited the neuromuscular modulation in those areas. Although MF, which reflects conduction velocity and fatigue progression, did not significantly differ between conditions, the relative stability of MF after the fatiguing task may reflect attenuation of performance fatigability-related decline [11]. In summary, functional compression tights may selectively delay performance fatigability in stabilizing muscles like the VMO, but localized effects depend on anatomical and garment-specific factors.

### Effects of compression tights on cognitive performance fatigability

EEG activity in the frontal regions (Fp1, Fp2, and Fz), associated with motor and cognitive performance fatigability, was analyzed to assess the effects of functional compression tights. Power changes in alpha, beta, and theta bands were used as neural indices of cognitive fatigability, reflecting relaxation (alpha), attention and cognitive effort (beta) [28,29], and cognitive fatigability (theta) [17]. While no statistically significant differences were observed between conditions, alpha power at Fz showed a moderate effect size. This suggests that compression tights may stabilize alpha activity in the Fz region, which is involved in motor control and fatigue-related attentional regulation. Such stabilization may indicate an attenuation of neurophysiological fatigability-related responses under physical stress. These findings imply that compression tights could modulate cognitive performance fatigability through effects on frontal brain dynamics.

### Interaction among perceived fatigability, muscle activity, and cortical responses

This study explored the interactions among muscle activity, EEG dynamics, and perceived fatigability under the influence of functional compression tights. Under the compression condition, the RMS of the RF positively correlated with power changes in the alpha, beta, and theta bands at Fp2 and Fz, while ST RMS showed a positive correlation with Fz-beta power. These findings suggest that functional compression tights may facilitate coordinated neuromuscular and cortical responses by stabilizing frontal brain activity, potentially enhancing motor performance during motor fatigability. In contrast, without compression tights, the RMS of the VMO showed significant negative correlations with alpha, beta, and theta power changes in the same regions, indicating less efficient neuromuscular regulation. Furthermore, reduced perceived fatigability perception under the compression condition was associated with lower RMS and attenuated MF reduction in the RF, implying suppression of excessive muscular load and maintenance of muscle fiber conduction velocity [11]. In terms of cortical indicators, significant associations were observed between perceived fatigability and alpha

and theta power at Fp2 and Fz under the compression condition, suggesting stable neural activity. By contrast, without compression tights, stronger and broader associations—including beta power—were evident, reflecting elevated cognitive strain [30]. These results are consistent with previous studies showing that compression garments reduce muscle fatigue and improve neuromuscular performance by enhancing blood flow and muscle stability [7,12,25]. Collectively, the findings suggest that functional compression tights may exert their fatigability-attenuating effects through two complementary physiological mechanisms: (1) optimizing local muscle activity and (2) attenuating central fatigability-related responses, as reflected in EEG activity.

## Clinical and functional implications of neurophysiological findings

The novelty of this study lies in its integrative evaluation of the fatigability-attenuating effects of functional compression tights from the perspectives of both motor and cognitive performance fatigability. Specifically, we combined peripheral muscle activity indicators (RMS and MF), reflecting motor performance fatigability, with frontal EEG measures (alpha, beta, and theta power), which serve as indices of cognitive performance fatigability, to assess the neurophysiological effects of wearing functional compression tights. From a neurophysiological standpoint, significant correlations were observed under with functional compression tights condition between perceived fatigability and alpha and theta power changes at Fp2 and Fz. These findings suggest that functional compression tights may reduce cognitive performance fatigability by stabilizing frontal neural activity and attenuating excessive EEG fluctuations. Additionally, correlations between suppressed activity in specific muscle groups (e.g., the VMO) and frontal EEG indices imply that functional compression tights may reduce unnecessary peripheral muscle activity, contributing to neurophysiological stability. Although no significant group-level differences were found in the Fz EEG power, moderate effect sizes were observed for the change rates in alpha power, which may indicate a stabilizing effect on frontal neural activity under physical stress. These findings suggest that wearing functional compression tights contributes to a significant reduction in perceived fatigability, efficient regulation of muscle activity, and attenuation of frontal EEG variability related to fatigue. From a clinical perspective, these results support the potential of functional compression tights as assistive tools in rehabilitation and exercise therapy, particularly for reducing motor performance fatigability and enhancing neuromuscular and central nervous system coordination to improve motor performance. In sports medicine, functional compression tights may also contribute to injury prevention by mitigating fatigability and can be integrated into comprehensive fatigue management strategies, potentially in combination with other wearable garments or monitoring technologies, for practical use in clinical and athletic settings.

## Limitations and future directions

This study has several limitations that should be acknowledged. First, although the knee joint angle was standardized during squatting, the trunk and hip joint angles were not controlled. Variations in these postures may have influenced muscle activation patterns and fatigability responses. Second, the tights were worn prior to the fatiguing task, which may have affected baseline muscle activity or perceived fatigability due to mechanical or psychological factors, including potential placebo effects. Third, the EMG analysis focused on selected lower limb muscles, such as the RF, VMO, and VL. Other relevant muscles involved in squatting, such as the gastrocnemius and erector spinae, were not assessed. Future studies should incorporate broader muscle group analyses and consider controlling joint kinematics more comprehensively. In addition, the integration of biomechanical and validated psychological assessments may help clarify the underlying mechanisms of compression garments and optimize their use in both athletic and clinical contexts.

## Conclusion

This study investigated the effects of functional compression tights on both perceived and performance fatigability. Compared to the without functional compression tights condition, wearing functional compression tights significantly reduced

perceived fatigability. It influenced fatigue-related suppression of muscle activity in specific muscles and emotion-related EEG indicators in the frontal cortex. These findings suggest that functional compression tights may exert fatigue-reducing effects through a dual mechanism, modulating peripheral muscle and central nervous activities, particularly in the context of emotional and attentional neural responses.

## Supporting information

**S1 Data.  Electromyography data used in the study.**
(XLSX)

**S2 Data.  Electroencephalography data used in the study.**
(XLSX)

**S3 Data.  Subjective fatigue scores recorded by NRS.**
(XLSX)

## Acknowledgments

We gratefully acknowledge the cooperation of Wacoal Corp. for their technical assistance and provision of study materials.

## Author contributions

**Conceptualization:** Takuma Isshiki, Shinnosuke Tsukada, Hayato Shigetoh, Junya Miyazaki.

**Data curation:** Takuma Isshiki, Shinnosuke Tsukada.

**Formal analysis:** Takuma Isshiki, Shinnosuke Tsukada, Hayato Shigetoh, Takayuki Kodama.

**Funding acquisition:** Junya Miyazaki.

**Investigation:** Takuma Isshiki, Shinnosuke Tsukada, Minami Akao, Yuna Ishikura, Hayato Shigetoh.

**Methodology:** Takuma Isshiki, Shinnosuke Tsukada, Hayato Shigetoh, Takayuki Kodama.

**Project administration:** Hayato Shigetoh, Junya Miyazaki.

**Resources:** Hayato Shigetoh.

**Software:** Hayato Shigetoh.

**Supervision:** Hayato Shigetoh, Takayuki Kodama, Junya Miyazaki.

**Validation:** Hayato Shigetoh.

**Visualization:** Takuma Isshiki, Shinnosuke Tsukada, Hayato Shigetoh.

**Writing – original draft:** Takuma Isshiki, Shinnosuke Tsukada.

**Writing – review & editing:** Minami Akao, Yuna Ishikura, Hayato Shigetoh, Takayuki Kodama, Junya Miyazaki.

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
