## [Decision Letter · Decision Letter 0]

16 Jun 2025

Dear Dr. Shigetoh,

Thank you for submitting your manuscript to PLOS ONE. After careful consideration, we feel that it has merit but does not fully meet PLOS ONE’s publication criteria as it currently stands. Therefore, we invite you to submit a revised version of the manuscript that addresses the points raised during the review process.

We look forward to receiving your revised manuscript.

Kind regards,

Hasan Sozen

Academic Editor

PLOS ONE

 [This research was supported by a collaborative research grant from Wacoal Corp. and by a Grant-in-Aid for Scientific Research (KAKENHI) from the Japan Society for the Promotion of Science (JSPS) [Grant Number 25K141790].]. 

3.  Please upload a copy of Figure 3 to 8, to which you refer in your text on page 16 and 20. If the figure is no longer to be included as part of the submission please remove all reference to it within the text.

Reviewers' comments:

Reviewer's Responses to Questions

**Comments to the Author**

1. Is the manuscript technically sound, and do the data support the conclusions?

Reviewer #1: Yes

Reviewer #2: Partly

2. Has the statistical analysis been performed appropriately and rigorously?

Reviewer #1: Yes

Reviewer #2: Yes

3. Have the authors made all data underlying the findings in their manuscript fully available?

Reviewer #1: Yes

Reviewer #2: Yes

4. Is the manuscript presented in an intelligible fashion and written in standard English?

Reviewer #1: Yes

Reviewer #2: No

Reviewer #1: Dear authors,

Greetings.

The paper presents a solid foundation to support the objectives and hypothesis. The hypothesis was well formulated and adequately addressed.

However, from my point of view, in the section:

"Some studies have reported limited supportive effects of compression tights or raised concerns regarding movement restriction or unnatural changes in muscle activation [4]. Moreover, many existing studies are limited to short-term evaluations, and long-term effects and safety have not been thoroughly examined [5]."

More studies could be cited.

Additionally, the statements from lines 88 to 97, for instance, require proper citations. In general, it is important to avoid claims without references.

The statistical analysis was well conducted, using both parametric and non-parametric methods. Indeed, whenever possible, subjective data could be transformed into numerical information.

The conclusions are supported by the results.

Based on the considerations above, I believe that this article deserves to be published after Minor Revisions.

Reviewer #2: Thank you for the opportunity to review this article. This study addresses an interesting topic, namely the effects of functional compressive garments of fatigability during squat exercises. However, in my opinion, the manuscript is not yet suitable for publication.

Here are some suggestions on how to improve the manuscript:

1. This investigation examined the effects of functional compressive garments on muscular performance during a fatiguing protocol involving five muscles of the dominant lower extremity. A critical limitation in the interpretation of findings concerns the predominantly null results observed across four of the five examined muscles. The selective positive outcome in the vastus medialis, contrasted with the absence of significant effects in the vastus lateralis, warrants more thorough examination considering the biomechanical contributions of these muscles during squatting movements. The authors should clarify whether the measurements pertained specifically to the vastus medialis obliquus, as this distinction carries important functional implications. The emphasis placed on the singular positive finding in the vastus medialis appears disproportionate relative to the null findings, which constitute the majority of the observed outcomes. A more comprehensive discussion of the negative results is warranted, particularly given that isolated muscle-specific positive effects represent a recurring pattern in compression garment research that merits critical evaluation.

2. Figures 3, 4, 5, 6, 7, 8 and 9 are missing.

3. References need to be formatted in a uniquely defined style.

4. The introduction and discussion sections are too long. For example, please delete lines 347-358 as this is a repetition of the results. Also, delete lines 402-405 as these concepts have already been mentioned earlier in the discussion. Also, add subheadings in the discussion.

5. Please be consistent throughout the manuscript with the terms “functional compression tights” and “functional tights”. These terms are currently used interchangeably, but the meaning is slightly different.

6. Similarly to point 4. It should be borne in mind that fatigue and muscle activity, although being interrelated phenomena, are distinct concepts. The authors use both terms in relation to the two parameters extracted from the EMG signal (MF and RMS), being sometimes indices of muscle activity and sometimes indices of muscle fatigue.

7. References 6, 7, 8 and 9 are obsolete. Consequently, the definition of fatigue used by the authors is obsolete. Please update to the new taxonomy initially proposed by Enoka and Duchateau (2016) and subsequently by Behrens et al. (2023). Replace the concepts of physical and mental fatigue with those of performance and perceived fatigability.

8. Many recent studies on the effects of compression garments on fatigue have not been cited (e.g. Wang and Li, 2022; Bajelani et al., 2021 and 2022; Belbasis and Fuss, 2018). Please update the bibliography.

9. Add p-values to the correlation analyses, in the results section. For example, the authors state in line 321 that a ‘significant positive correlation’ was found, but the p-value is missing.

10. The characterization of p=0.05 as representing a "trend toward significance" requires correction. As demonstrated by Nead et al. (2018), "There is no definition of a trend toward statistical significance and, therefore, describing 'almost significant' results as a trend introduces substantial subjectivity" and constitutes biased reporting practices. Such terminology lacks methodological rigor and may mislead readers regarding statistical evidence. To enhance precision and transparency in reporting, the authors should present p-values to three decimal places rather than rounding to two. This modification would provide readers with more accurate information for interpreting the statistical evidence and eliminate ambiguous language that undermines scientific objectivity.

MINOR COMMENTS

Title: I would add “during squat exercises”.

L91: add abbreviation of mean frequency

L111: add definition of the three frontal regions Fp1, Fp2, Fz. Actually, the definition appears to be at line 401.

L147 No patients were recruited in this study. Please modify.

L164 add city and state

L200 add reference for SENIAM

L208-9 see comment above

L225-8 Not clear. Please rephrase

L274 add “only” before the vastus medialis

L289 why the result on the Fz site is considered as “marginally significant trend” (p=0.05), whereas in the previous paragraph the result of RMS in BF was not (p=0.05)?

**Do you want your identity to be public for this peer review?** For information about this choice, including consent withdrawal, please see our Privacy Policy

Reviewer #1: **Yes: ** Tales Alexandre Aversi Ferreira

Reviewer #2: No

---

## [Author Response · Author response to Decision Letter 1]

17 Jul 2025

17-July-2025

Dr. Emily Chenette

Editor-in-Chief

Dr. Hasan Sozen

Academic Editor

PLOS ONE

Manuscript for submission: "Evaluation of the fatigue reduction by functional compression tights during squat exercise via electromyography and electroencephalography analysis" (PONE-D-25-23745)

Dear Editor,

We would like to express our sincere appreciation for the valuable comments and suggestions provided by the reviewers regarding our manuscript entitled:

“Evaluation of the fatigue reduction by functional compression tights during squat exercises via electromyography and electroencephalography analysis” (PONE-D-25-23745).

We have carefully addressed all reviewer comments and revised the manuscript accordingly. Major changes include:

Updating outdated references and incorporating more recent studies related to compression garments and fatigue.

Refining the conceptual framework of fatigue based on the taxonomy proposed by Enoka and Duchateau (2016) and Behrens et al. (2023).

Revising the discussion of muscle-specific effects, especially regarding the vastus medialis obliquus.

Correcting inconsistencies in p-value reporting and eliminating ambiguous statistical terms.

Improving clarity and conciseness by removing redundant statements and standardizing terminology throughout the manuscript.

All revisions in the manuscript are highlighted in red. We have also updated the reference format to comply with the Vancouver style as per journal requirements.

We sincerely thank you and the reviewers again for the thoughtful feedback, which helped us to substantially improve the quality and clarity of our manuscript. We hope that the revised version will now be acceptable for publication in PLOS ONE.

Sincerely,

Hayato Shigetoh, PhD

Department of Physical Therapy

Faculty of Health Science

Kyoto Tachibana University

Yamashina-ku, Kyoto, Japan

Tel.: +81-75-571-1111

Email: shigeto@tachibana-u.ac.jp

Reviewer #1:

Reviewer comments:

Dear authors,

Greetings.

The paper presents a solid foundation to support the objectives and hypothesis. The hypothesis was well formulated and adequately addressed.

However, from my point of view, in the section:

Response:

Thank you for reviewing our manuscript. Your comments were highly insightful and enabled us to greatly improve the quality of our manuscript. In the following pages are our point-by-point responses to each of the comments of the reviewers as well as your own comments. Revisions in the text are shown using red-colored font. Also, the paper has been edited and proofread by a professional native English speaking editing service.

Reviewer comments:

"Some studies have reported limited supportive effects of compression tights or raised concerns regarding movement restriction or unnatural changes in muscle activation [4]. Moreover, many existing studies are limited to short-term evaluations, and long-term effects and safety have not been thoroughly examined [5]."

More studies could be cited.

Response:

Thank you for your suggestion. In response to your comment, we have added additional references to support the statements regarding the limited effects of compression tights, potential concerns about movement restriction, and the lack of long-term evaluations.

We have revised the following text:

Line 60: Some studies have reported inconsistent benefits or even adverse effects such as restricted movement or altered muscle activation [4–6]. Additionally, most research focuses on short-term effects, and the long-term efficacy and safety of these garments remain unclear [7,8].

Reviewer comments:

Additionally, the statements from lines 88 to 97, for instance, require proper citations. In general, it is important to avoid claims without references.

Response:

Thank you for your valuable comment regarding the use of EMG indicators for muscle fatigue assessment. In response, we have revised the relevant paragraph in the manuscript to clearly distinguish between time-domain and frequency-domain measures derived from EMG signals. Specifically, we now clarify that root mean square (RMS) is a time-domain indicator reflecting the amplitude of muscle activation, which may increase during early stages of fatigue due to greater motor unit recruitment, but may also decline under severe fatigue due to diminished contractile capacity. In contrast, mean frequency (MF) is a frequency-domain indicator that typically decreases during fatigue due to reduced muscle fiber conduction velocity. To support this explanation, we have added a citation We have revised the following text:

Line 75: Muscle fatigue can be objectively quantified using surface electromyography (EMG), which captures myoelectric changes in both time and frequency domains. In the time domain, root mean square (RMS) reflects the amplitude of muscle activation, typically increasing in early fatigue due to greater motor unit recruitment, then decreasing under severe fatigue. In the frequency domain, mean frequency (MF) generally decreases with progressive fatigue as muscle fiber conduction velocity slows. RMS and MF are therefore considered complementary indicators of muscle fatigue [11].

Reviewer comments:

The statistical analysis was well conducted, using both parametric and non-parametric methods. Indeed, whenever possible, subjective data could be transformed into numerical information.

The conclusions are supported by the results.

Based on the considerations above, I believe that this article deserves to be published after Minor Revisions.

Response:

Thank you very much for your valuable comments and suggestions. We have carefully addressed all of your points and revised the manuscript accordingly. Your feedback has been extremely helpful in improving the clarity, accuracy, and overall quality of the paper. We sincerely appreciate your contribution to enhancing our work.

Reviewer #2:

Reviewer comments:

Thank you for the opportunity to review this article. This study addresses an interesting topic, namely the effects of functional compressive garments of fatigability during squat exercises. However, in my opinion, the manuscript is not yet suitable for publication.

Here are some suggestions on how to improve the manuscript:

Response:

Thank you for reviewing our manuscript. Your comments were highly insightful and enabled us to greatly improve the quality of our manuscript. In the following pages are our point-by-point responses to each of the comments of the reviewers as well as your own comments. Revisions in the text are shown using red-colored font. Also, the paper has been edited and proofread by a professional native English speaking editing service.

Reviewer comments:

1. This investigation examined the effects of functional compressive garments on muscular performance during a fatiguing protocol involving five muscles of the dominant lower extremity. A critical limitation in the interpretation of findings concerns the predominantly null results observed across four of the five examined muscles. The selective positive outcome in the vastus medialis, contrasted with the absence of significant effects in the vastus lateralis, warrants more thorough examination considering the biomechanical contributions of these muscles during squatting movements. The authors should clarify whether the measurements pertained specifically to the vastus medialis obliquus, as this distinction carries important functional implications. The emphasis placed on the singular positive finding in the vastus medialis appears disproportionate relative to the null findings, which constitute the majority of the observed outcomes. A more comprehensive discussion of the negative results is warranted, particularly given that isolated muscle-specific positive effects represent a recurring pattern in compression garment research that merits critical evaluation.

Response:

Thank you for your insightful and constructive feedback. We acknowledge the importance of contextualizing the selective results observed in the vastus medialis.

In our study, surface EMG electrodes were specifically placed on the vastus medialis obliquus (VMO), and we have revised the manuscript accordingly to clarify this anatomical specificity. The VMO plays a crucial role in patellar stabilization, especially during squatting movements, and may respond differently to external interventions than other quadriceps muscles (Powers, 2000).

We agree that the interpretation of isolated findings requires caution. However, previous studies have reported that compression garments can suppress muscle activity depending on the compression pressure and region applied (Leabeater et al., 2022; Broatch et al., 2020). Since the VMO region may receive distinct pressure profiles from functional compression tights, regional differences in pressure could contribute to the observed variation in EMG responses across muscles.

Moreover, the orientation of the compression bands relative to the muscle fiber direction has been shown to influence EMG activity (Chaudhari et al., 2014). Therefore, the configuration of the functional compression tights used in our study may have contributed to muscle-specific effects.

Importantly, the absence of significant changes in muscles other than the VMO may also be explained by such regional variations in compression intensity and the directionality of compression bands. These garment design factors likely result in differential mechanical and neuromuscular effects across muscle groups, thereby limiting the uniformity of compression-induced changes.

We have added a more balanced discussion in the revised manuscript to reflect both the observed null findings and the potential mechanisms underlying the isolated result in the VMO.

We have revised the following text:

Line 331: Wearing functional compression tights reduced subjective lower limb fatigue and selectively suppressed EMG activity in the vastus medialis obliquus (VMO) after a fatiguing squat task. Since RMS typically increases with fatigue [11], the lower RMS observed in the VMO suggests reduced neuromuscular demand. This may be due to enhanced joint stability, as the VMO plays a key role in patellar tracking [24]. Prior studies have shown that compression garments can modulate muscle activity based on regional pressure and the alignment of compression bands with muscle fiber direction [25-27]. These factors may explain why no significant effects were found in other muscles like the rectus femoris or semitendinosus. Regional variation in compression strength and orientation may have limited the neuromuscular modulation in those areas. Although median frequency (MF), which reflects conduction velocity and fatigue progression, did not differ significantly, a trend toward preserved MF suggests attenuated fatigue-related decline [11]. In summary, functional compression tights may selectively delay fatigue in stabilizing muscles like the VMO, but localized effects depend on anatomical and garment-specific factors.

Reviewer comments:

2. Figures 3, 4, 5, 6, 7, 8 and 9 are missing.

Response:

Thank you for pointing this out. We apologize for the oversight. Due to an upload error during the initial submission, Figures 3 through 9 were not properly included. We have now uploaded all missing figures as part of the revised submission. We appreciate your understanding.

Reviewer comments:

3. References need to be formatted in a uniquely defined style.

Response:

Thank you for your comment. In response, we have carefully revised the reference list to conform to the journal’s required formatting style. All references have been reviewed and reformatted accordingly to ensure consistency and compliance.

Reviewer comments:

4. The introduction and discussion sections are too long. For example, please delete lines 347-358 as this is a repetition of the results. Also, delete lines 402-405 as these concepts have already been mentioned earlier in the discussion. Also, add subheadings in the discussion.

Response:

Thank you for your valuable feedback. In response to your suggestion, we revised the manuscript by deleting redundant content in the Introduction and Discussion sections. Specifically, lines 347–358, which repeated the results, and lines 402–405, which overlapped conceptually with earlier text, were removed. Furthermore, we added subheadings to the Discussion to improve clarity and structure. We have carefully reviewed the entire manuscript to avoid unnecessary repetition and ensure conciseness.

Reviewer comments:

5. Please be consistent throughout the manuscript with the terms “functional compression tights” and “functional tights”. These terms are currently used interchangeably, but the meaning is slightly different.

Response:

Thank you for your helpful comment. In response, we have revised the manuscript to ensure consistent use of terminology. Specifically, we have replaced all instances of “functional tights” with “functional compression tights” to accurately reflect the characteristics of the garment used in this study and to avoid potential confusion. This terminology has been standardized throughout the manuscript.

Reviewer comments:

6. Similarly to point 4. It should be borne in mind that fatigue and muscle activity, although being interrelated phenomena, are distinct concepts. The authors use both terms in relation to the two parameters extracted from the EMG signal (MF and RMS), being sometimes indices of muscle activity and sometimes indices of muscle fatigue.

Response:

Thank you for this important comment. We fully agree that muscle activity and muscle fatigue are distinct, though interrelated, physiological concepts. In response, we have carefully revised the manuscript to ensure consistent and precise use of terminology.

Specifically, we now clarify that root mean square (RMS) is a time-domain indicator that primarily reflects muscle activation, and may increase during fatigue due to enhanced motor unit recruitment and synchronization. In contrast, mean frequency (MF) is a frequency-domain indicator that reflects muscle fatigue, typically decreasing as muscle fiber conduction velocity declines. Therefore, while both RMS and MF are derived from the EMG signal, they represent complementary yet distinct aspects of muscle function during fatiguing tasks—RMS indicating temporal amplitude changes related to activation, and MF indicating spectral shifts associated with fatigue.

We have revised the following text:

Line 75: Muscle fatigue can be objectively quantified using surface electromyography (EMG), which captures myoelectric changes in both time and frequency domains. In the time domain, root mean square (RMS) reflects the amplitude of muscle activation, typically increasing in early fatigue due to greater motor unit recruitment, then decreasing under severe fatigue. In the frequency domain, mean frequency (MF) generally decreases with progressive fatigue as muscle fiber conduction velocity slows. RMS and MF are therefore considered complementary indicators of muscle fatigue [11].

Reviewer comments:

7. References 6, 7, 8 and 9 are obsolete. Consequently, the definition of fatigue used by the authors is obsolete. Please update to the new taxonomy initially proposed by Enoka and Duchateau (2016) and subsequently by Behrens et al. (2023). Replace the concepts of physical and mental fatigue with those of performance and perceived fatigability.

Response:

Thank you very much for this insightful comment. In response, we have removed the outdated description of fatigue based on the traditional classification into "physical" and "mental" fatigue. Instead, we have updated the conceptual framework of fatigue in accordance with the taxonomy proposed by Enoka and Duchateau (2016) and further elaborated by Behrens et al. (2023).

Specifically, we now define fatigue as a psychophysiological condition characterized by two interrelated components: performance fatigability (a measurable decline in task performance) and perceived fatigability (a subjective sensation of fatigue). Furthermore, we clarified that performance fatigability can be subdivided into motor performance fatigability and cognitive performance fatigability, depending on the nature of

---

## [Decision Letter · Decision Letter 1]

28 Jul 2025

Dear Dr. Shigetoh,

Thank you for submitting your manuscript to PLOS ONE. After careful consideration, we feel that it has merit but does not fully meet PLOS ONE’s publication criteria as it currently stands. Therefore, we invite you to submit a revised version of the manuscript that addresses the points raised during the review process.

We look forward to receiving your revised manuscript.

Kind regards,

Hasan Sozen

Academic Editor

PLOS ONE

Journal Requirements:

Reviewers' comments:

Reviewer's Responses to Questions

**Comments to the Author**

Reviewer #1: All comments have been addressed

Reviewer #2: (No Response)

2. Is the manuscript technically sound, and do the data support the conclusions?

Reviewer #1: Yes

Reviewer #2: Yes

3. Has the statistical analysis been performed appropriately and rigorously?

Reviewer #1: Yes

Reviewer #2: Yes

4. Have the authors made all data underlying the findings in their manuscript fully available?

Reviewer #1: Yes

Reviewer #2: Yes

5. Is the manuscript presented in an intelligible fashion and written in standard English?

Reviewer #1: Yes

Reviewer #2: Yes

Reviewer #1: All of my requested revisions were considered. The authors increased the number of references and edited and improved some paragraphs.

Reviewer #2: I would like to thank the authors for answering most of my questions and suggestions for improving their manuscript. There are still a few points that need to be fixed so that the manuscript can then be accepted for publication.

1. During my initial revision, I suggested that the authors update the definition of fatigue to that proposed by Duchateau and Enoka 10 years ago. However, the authors did not make corrections throughout the manuscript, so in the current form there is a lot of confusion around the term fatigue and fatigability, because both terms are used. For example, why did the authors not change the term “subjective fatigue” to the more correct and accurate term “perceived fatigability”? For example, what is “subjective lower limb fatigue”?

I again suggest that the authors be consistent with the terminology throughout the manuscript, including the abstract!

2. p-values

Once again, I strongly suggest that the authors use three decimal places in all results. For example, in Table 2, the decimal p-values are still limited to two. Please amend.

3. The issue about “trend toward significance”, which I pointed out in the first revision phase, has not been completely resolved. Please modify at lines 350-352.

4. Muscle terminology

The authors changed the term “vastus medialis” to the more precise “vastus medialis obliquus”. However, this has not been done throughout the manuscript. Please amend accordingly.

5. Figures

Many figures still contain Japanese terms. Please change them to English. Furthermore, the figures appear too small to be observed correctly by readers. Please edit.

Minor comments

L138, L149, L150, etc “a muscle fatigue task”  a fatiguing task.

L184 Time  time

L281 please add a space between “the” and “condition”.

**Do you want your identity to be public for this peer review?** For information about this choice, including consent withdrawal, please see our Privacy Policy

Reviewer #1: **Yes: ** Tales Alexandre Aversi-Ferreira

Reviewer #2: No

---

## [Author Response · Author response to Decision Letter 2]

29 Jul 2025

29-July-2025

Dr. Emily Chenette

Editor-in-Chief

Dr. Hasan Sozen

Academic Editor

PLOS ONE

Manuscript for submission: "Evaluation of the reduction in perceived and performance fatigability by functional compression tights during squat exercise via electromyography and electroencephalography analysis" (PONE-D-25-23745)

Dear Editor,

We would like to express our sincere appreciation for the valuable comments and suggestions provided by the reviewers regarding our manuscript entitled:

“Evaluation of the reduction in perceived and performance fatigability by functional compression tights during squat exercise via electromyography and electroencephalography analysis” (PONE-D-25-23745).

In response to the reviewer comments, we have thoroughly revised the manuscript. Major modifications include:

Clarification and standardization of terminology related to fatigue and fatigability throughout the manuscript, including the abstract, based on the taxonomy proposed by Enoka and Duchateau (2016).

Replacement of the term “subjective fatigue” with the more accurate expression “perceived fatigability” where appropriate.

Correction of p-value formatting, ensuring consistency to three decimal places across the abstract, results, tables, and figure legends.

Removal of all instances of ambiguous statistical phrasing such as “trend toward significance,” in line with reviewer recommendations.

Consistent use of the term “vastus medialis obliquus (VMO)” throughout the text.

Complete revision of all figures to ensure English-only labeling and improved font sizes for readability. All revisions in the manuscript are highlighted in red. We have also updated the reference format to comply with the Vancouver style as per journal requirements.

All revisions are clearly marked using Track Changes in the manuscript.

We are grateful for the reviewers’ and editors’ comments, which have greatly improved the clarity and scientific rigor of our manuscript. We sincerely hope that the revised version is now suitable for publication in PLOS ONE.

Sincerely,

Hayato Shigetoh, PhD

Department of Physical Therapy

Faculty of Health Science

Kyoto Tachibana University

Yamashina-ku, Kyoto, Japan

Tel.: +81-75-571-1111

Email: shigeto@tachibana-u.ac.jp

Reviewer #1:

Reviewer comments:

All of my requested revisions were considered. The authors increased the number of references and edited and improved some paragraphs.

Response:

Thank you very much for your constructive feedback and thoughtful comments throughout the review process. We are grateful that all of your requested revisions were considered acceptable. Your suggestions have contributed significantly to improving the clarity, quality, and overall rigor of the manuscript. We believe that your comments have helped us develop a stronger and more impactful paper.

Reviewer #2:

Reviewer comments:

I would like to thank the authors for answering most of my questions and suggestions for improving their manuscript. There are still a few points that need to be fixed so that the manuscript can then be accepted for publication.

Response:

Thank you for reviewing our manuscript. Your comments were highly insightful and enabled us to greatly improve the quality of our manuscript. In the following pages are our point-by-point responses to each of the comments of the reviewers as well as your own comments. Revisions in the text are shown using Track Changes. Also, the paper has been edited and proofread by a professional native English speaking editing service.

Reviewer comments:

1. During my initial revision, I suggested that the authors update the definition of fatigue to that proposed by Duchateau and Enoka 10 years ago. However, the authors did not make corrections throughout the manuscript, so in the current form there is a lot of confusion around the term fatigue and fatigability, because both terms are used.

For example, why did the authors not change the term “subjective fatigue” to the more correct and accurate term “perceived fatigability”? For example, what is “subjective lower limb fatigue”?

I again suggest that the authors be consistent with the terminology throughout the manuscript, including the abstract!

Response:

Thank you for your valuable comment regarding the terminology of fatigue and fatigability. In response to your suggestion, we carefully reviewed the manuscript and revised the terminology throughout to ensure consistency and accuracy based on the definition proposed by Duchateau and Enoka. Specifically, we introduced the definition of fatigability in the Introduction and subsequently replaced terms such as “subjective fatigue” with “perceived fatigability” where appropriate, including in the Abstract, Methods, Results, Discussion, and Conclusion. We appreciate your guidance, which helped us improve the conceptual clarity of the manuscript.

Reviewer comments:

2. p-values

Once again, I strongly suggest that the authors use three decimal places in all results. For example, in Table 2, the decimal p-values are still limited to two. Please amend.

Response:

Thank you for pointing this out again. In accordance with your suggestion, we have revised all p-values throughout the manuscript—including those in the Abstract, main text, tables, and figure legends—to consistently report values to three decimal places. We carefully re-checked each occurrence to ensure uniformity and precision in statistical reporting. We appreciate your attention to detail, which has contributed to the improvement of the manuscript’s clarity and scientific rigor.

Reviewer comments:

3. The issue about “trend toward significance”, which I pointed out in the first revision phase, has not been completely resolved. Please modify at lines 350-352.

Response:

Thank you for your insightful comment. In response, we thoroughly reviewed the manuscript, including the Abstract, Results, and Discussion sections, and removed or rephrased all instances of the expression “trend toward significance” as appropriate. These changes were made to ensure clarity and consistency in statistical reporting, and to avoid potentially ambiguous interpretations. We appreciate your guidance in improving the quality of the manuscript.

Reviewer comments:

4. Muscle terminology

The authors changed the term “vastus medialis” to the more precise “vastus medialis obliquus”. However, this has not been done throughout the manuscript. Please amend accordingly.

Response:

Thank you for your valuable suggestion. In accordance with your comment, we have carefully reviewed the entire manuscript and consistently replaced the term “vastus medialis” with the more precise term “vastus medialis obliquus” or its abbreviation “VMO” where appropriate. This ensures anatomical accuracy and clarity throughout the manuscript.

Reviewer comments:

5. Figures

Many figures still contain Japanese terms. Please change them to English.

Furthermore, the figures appear too small to be observed correctly by readers. Please edit.

Response:

Thank you for your helpful comment. In response, we thoroughly reviewed all figures to ensure that no Japanese terms remain and confirmed that all labels and annotations are now fully presented in English. Additionally, we have adjusted the font sizes and overall layout of each figure to improve visibility and clarity, ensuring they can be easily interpreted by readers. All revised figures have been resubmitted accordingly.

Reviewer comments:

Minor comments

L138, L149, L150, etc “a muscle fatigue task”  a fatiguing task.

Response:

I have revised it.

Reviewer comments:

L184 Time  time

Response:

I have revised it.

Reviewer comments:

L281 please add a space between “the” and “condition”.

Response:

I have revised it.

---

## [Editor Report · Decision Letter 2]

21 Aug 2025

Evaluation of the reduction in perceived and performance fatigability by functional compression tights during squat exercises via electromyography and electroencephalography analysis

PONE-D-25-23745R2

Dear Dr. Shigetoh,

We’re pleased to inform you that your manuscript has been judged scientifically suitable for publication and will be formally accepted for publication once it meets all outstanding technical requirements.

Kind regards,

Hasan Sozen

Academic Editor

PLOS ONE
---

## [Editor Report · Acceptance letter]

PONE-D-25-23745R2

PLOS ONE

Dear Dr. Shigetoh,

I'm pleased to inform you that your manuscript has been deemed suitable for publication in PLOS ONE. Congratulations! Your manuscript is now being handed over to our production team.

Kind regards,

on behalf of

Assoc. Prof. Hasan Sozen

Academic Editor

PLOS ONE